# A Partially Protective Vaccine for *Fasciola hepatica* Induced Degeneration of Adult Flukes Associated to a Severe Granulomatous Reaction in Sheep

**DOI:** 10.3390/ani11102869

**Published:** 2021-09-30

**Authors:** Verónica Molina-Hernández, María T. Ruiz-Campillo, Francisco J. Martínez-Moreno, Leandro Buffoni, Álvaro Martínez-Moreno, Rafael Zafra, María J. Bautista, Alejandro Escamilla, Raúl Pérez-Caballero, José Pérez

**Affiliations:** 1Departamento de Anatomía y Anatomía Patológica Comparadas y Toxicología, Facultad de Veterinaria, Universidad de Córdoba, Edificio de Sanidad Animal, Campus de Rabanales, Ctra. Madrid-Cádiz Km 396, 14014 Córdoba, Spain; b62mohev@uco.es (V.M.-H.); v42rucam@uco.es (M.T.R.-C.); mjbautista@uco.es (M.J.B.); an1pearj@uco.es (J.P.); 2Departamento de Sanidad Animal (Parasitología), Facultad de Veterinaria, Universidad de Córdoba, Edificio de Sanidad Animal, Campus de Rabanales, Ctra. Madrid-Cádiz Km 396, 14014 Córdoba, Spain; h12bupel@uco.es (L.B.); amm@uco.es (Á.M.-M.); Rafael.zafra@uco.es (R.Z.); raulpc.vetpares@gmail.com (R.P.-C.); 3Departamento de Fisiología Humana, Histología Humana, Anatomía Patológica y Educación Físico Deportiva, Facultad de Medicina, Boulevard Louis Pasteur, 32, 29071 Málaga, Spain; jandromilla@uma.es

**Keywords:** *Fasciola hepatica*, vaccines, histopathology, immunohistochemistry, sheep

## Abstract

**Simple Summary:**

Fasciolosis is a parasitic disease of livestock causing important economic losses worldwide and it is also a zoonosis. Current therapy relies on the use of anthelmintic drugs, which is no longer sustainable due to the increase of anthelmintic resistance and the risk of drug residues in food. A deep understanding of the host-parasite interaction is required to develop protective vaccines for the control of fasciolosis. The aim of the present study is to evaluate the hepatic lesions in sheep vaccinated with a partly protective vaccine for *F. hepatica*, a non-protective vaccine and an infected control group. The protective vaccine showed less severe hepatic lesions than the infected control group. In addition, in the protective vaccine group dead flukes surrounded by a severe granulomatous inflammation were observed, which taken together with the lower fluke burden, suggests that the host response induced by the partially protective vaccine may have been involved in the death of adult flukes of *F. hepatica*. This is the first study reporting the presence of degenerated flukes associated to a severe granulomatous inflammation in bile ducts in a vaccine trial, a finding that would be useful for improving vaccine efficacy in future trials.

**Abstract:**

Fasciolosis is an important economic disease of livestock. There is a global interest in the development of protective vaccines since current anthelmintic therapy is no longer sustainable. A better knowledge of the host-parasite interaction is needed for the design of effective vaccines. The present study evaluates the microscopical hepatic lesions in sheep immunized with a partially protective vaccine (VAC1), a non-protective vaccine (VAC2), and an infected control group (IC). The nature of granulomatous inflammation associated with degeneration of adult flukes found in the VAC1 group was characterized by immunohistochemistry. Hepatic lesions (fibrous perihepatitis, chronic tracts, bile duct hyperplasia, infiltration of eosinophils and lymphocytes and plasma cells) were significantly less severe in the VAC1 group than in the IC group. Dead adult flukes within bile ducts were observed only in the VAC1 group and were surrounded by a severe granulomatous inflammation composed by macrophages and multinucleate giant cells with a high expression of lysozyme, CD163 and S100 markers, and a low expression of CD68. Numerous CD3+ T lymphocytes and scarce infiltrate of FoxP3+ Treg and CD208+ dendritic cells were present. This is the first report describing degenerated flukes associated to a severe granulomatous inflammation in bile ducts in a *F. hepatica* vaccine trial.

## 1. Introduction

Fasciolosis is a zoonosis caused by the helminth *F. hepatica* with a significant economic [1] public health importance all over the world and it is considered by the WHO as a re-emerging neglected tropical disease [2]. After the ingestion of the infective form called metacercariae, the pathogenesis of *F. hepatica* involves a pre-hepatic stage starting with the penetration of newly excysted juveniles (NEJs) through the host intestine wall and its migration within the peritoneal cavity. In the hepatic stage of pathogenesis, the NEJs reach and penetrate the liver capsule, and in the parenchyma the parasites move randomly, forming characteristic transects/tunnels and feed on the hepatic cellular components and blood making NEJs growing and developing rapidly. Lately, the parasites enter the bile ducts where they develop into their adult form and start releasing up to 20,000–24,000 eggs per fluke per day. The feeding and migratory activities cause tissue perforation meanwhile the presence of adult flukes within the bile ducts induces a severe chronic cholangitis with erosion and hyperplasia of the biliary epithelium leading to extensive tissue damage [3].

Nowadays, the control of the disease is based on the use of drugs, particularly triclabendazole, which is effective against multiple parasite stages. However, the growing resistance of the parasite to the chemical products and the concerns about chemical residues in food with their detrimental impact on the environment make the development of novel strategies critical that are more effective and sustainable. Thus, vaccines have been highlighted as the most suitable option to deal with the detrimental effects found with the current used chemical therapies [3,4].

However, over the last three decades there has been proposed numerous vaccine candidates and assessed in several animal models such as rats, mice, and rabbits with non-consistent results in ruminants [5,6]. The immunomodulatory capacity exerted by *F. hepatica* is claimed to be the main obstacle to produce an effective vaccine, some new strategies in vaccine development include the identification of protective peptides by mapping B-cell epitopes of immunodominant *F. hepatica* antigens, which has been recently reported in sheep [7] and cattle [8] with promising results.

Moreover, the assessment of vaccine protection is mainly based on parasitological and systemic immunological parameters. However, up to the date, the liver damage and local immune response is not a relevant parameter to assess the effectiveness of the vaccines against liver flukes and it could be a potent tool to complement the parasitological and serological studies carried out routinely in these vaccine trials. Furthermore, few studies have described the liver pathology in vaccine trials conducted in the natural hosts [9,10].

Based on these premises, we consider the reduction in the liver damage as a crucial feature to evaluate the vaccine effectiveness against *F. hepatica* taking into account that a reduction of the liver damage can be beneficial for the animal welfare and the improvement of the animal production, hence reducing the economic impact of the disease. The aim of the present study was to evaluate the microscopical hepatic damage from sheep immunized with a partly protective vaccine composed of four recombinant proteins from *F. hepatica* in adjuvant Montanide 61 VG, a non-protective vaccine composed of the same antigens in adjuvant Alhydrogel and an infected control group of sheep experimentally infected with *F. hepatica*. Granulomatous cholangitis associated to dead parasites found in sheep immunized with the protective vaccine were characterized using immunohistochemistry.

## 2. Materials and Methods

### 2.1. Experimental Design

Thirty-seven 8-month-old male Merino-breed sheep obtained from a liver fluke-free farm were used in this study. Before starting the experiment, the animals were treated with fenbendazole and subsequently confirmed to be negative for parasite eggs by fecal zinc-sulphate base flotation technique, with no eggs detected. Additionally, all animals were tested for serum IgG specific antibodies for *F. hepatica* cathepsin L1 (FhCL1) by ELISA, obtaining negative results in all cases. During the experience, animals were housed indoors in the experimental farm of the University of Cordoba and fed with hay and commercial pellet.

Sheep were randomly distributed in four groups called vaccine 1 (VAC1), vaccine 2 (VAC2), infected controls (IC) and uninfected controls (UC). Groups VAC1 (*n* = 10) and VAC2 (*n* = 10) were immunized subcutaneously with two doses, 4 weeks apart, of a multivalent vaccine. The formulation of the two vaccines assessed in this study were published previously by [11] finding a reduction of fluke burden (37.2%) and egg output (28.71%) in comparison to IC group. Briefly, each vaccine dose (2 mL) contained a cocktail of *F. hepatica* recombinant proteins including cathepsin L1 (rFhCL1), peroxiredoxin (rFhPrx), helminth defence molecules (rFhHDM), and leucine aminopeptidase (rFhLAP) at a concentration of 100 μg per antigen emulsified in two different adjuvants, Montanide ISA 61 VG (Seppic, Puteaux, France) and Alhydrogel^®^ adjuvant 2% (InvivoGen, San Diego, CA, USA), respectively. The *F. hepatica* recombinant proteins were obtained as described [11]. Group IC (*n* = 10) was unimmunized and infected; and group UC (*n* = 7) was unimmunized and uninfected. Eight weeks after the first immunization, groups VAC1, VAC2, and IC were infected orally with 150 metarcercariae of the South Gloucester strain of *F. hepatica* (Ridgeway Research Ltd., UK) administered in gelatin capsules with a dosing gun. At 15 weeks post-infection (wpi), all animals were culled in batches of six per day by intravenous injection of a proper dose of T61^®^ (MSD Animal Health, Salamanca, Spain) according to manufacturer’s instruction. The experiment was approved by the Bioethics Committee of the University of Cordoba (No. 1118, date 11 January 2016) and was performed considering European (2010/63/UE) and Spanish (L32/2007 and RD53/2013) directives on animal experimentation. Fluke burden and gross pathology of the liver were reported by [11].

### 2.2. Liver Pathology

During necropsy, livers were removed and a total of six hepatic samples per liver were collected from hepatic lesions from both the right and left hepatic lobes of each animal. In the UC group, liver samples were randomly collected from the left and right hepatic lobes. Next, all the samples were fixed in 10% buffered formalin for 24 h and routinely processed and embedded in paraffin wax. Tissue sections (4 µm thick) were stained with hematoxylin and eosin (H&E) and evaluated independently by two pathologists to assess the severity of the hepatic lesions per animal and group, as follows: 0, absent; 1, mild; 2, moderate; 3, severe; 4, very severe. The parameters scored were related to chronic stages of the infection and immunized groups (VAC1 and VAC2) were compared to the unimmunized and infected group (IC). The pathological changes studied were fibrous perihepatitis, chronic migratory tracts, bile duct hyperplasia, periportal fibrosis, granulomas, eosinophilic and lymphoplasmacytic infiltrates, globule leukocytes and parasite eggs within the bile ducts or in the hepatic parenchyma.

### 2.3. Immunohistochemistry

The avidin-biotin-peroxidase (ABC) method was used on paraffin wax liver sections of 3-μm thick as previously described [12]. Briefly, tissue sections were dewaxed, rehydrated and endogenous peroxidase activity was exhausted by incubation with 0.3% hydrogen peroxide in methanol for 30 min at room temperature (RT). Tissue sections were incubated in different retrieval antigen pre-treatments based on the primary antibody used (Table 1). After three rinses in phosphate-buffered saline (PBS, pH 7.2), tissue sections were incubated with 20% normal goat serum (MP Biomedicals) for 30 min at RT. Endogenous liver biotin was blocked using the Avidin/Biotin blocking kit (Vector Laboratories) following the manufacturer instructions. A panel of primary antibodies were diluted in PBS containing 10% normal goat serum (Table 1) and incubated overnight at 4 °C. Following washing in PBS, biotinylated goat anti-rabbit or anti-mouse secondary antibodies (Dako, Agilent, E0432 and E0433, respectively) diluted 1:200 and 1:50, respectively, were applied correspondingly for 30 min at RT. After washing in PBS, the sections were incubated with the ABC complex (Vectastain ABC Elite Kit) for 1 h at RT in darkness, washed in 0.05 M Tris buffered saline (pH 7.6), and then incubated in the chromogen solution (Vector NovaRED Peroxidase Substrate Kit). Finally, the sections were counterstained with Harris’ hematoxylin and mounted in Eukitt quick-hardening mounting medium. Tissue sections in which the specific primary antibodies were replaced by non-immune isotype antibody were used as negative controls.

### 2.4. Cell Counting

Immunostained cells with the different antibodies used in the study were counted in five areas of 0.08 µm^2^, randomly selected areas in granulomatous lesions associated to degenerated flukes. Results were expressed as mild: <10 immunostained cells per field; moderate: 10–30 immunostained cells per field; severe: 30–50 immunostained cells per field; and very severe: >50 immunostained cells per field.

### 2.5. Statistical Analysis

Statistical analysis was carried out for the histopathological hepatic lesions using the Graphpad Prism 7.0 software package (Graphpad Software, Inc., San Diego, CA, USA). A non-parametrical one-way ANOVA Kruskal–Wallis test with Dunn’s post-hoc test was carried out. *p* values < 0.01 were considered very statistically significant and *p* values < 0.05 were considered statistically significant.

## 3. Results

### 3.1. Histopathological Evaluation of Hepatic Lesions

The livers of the UC group showed no histopathological changes, portal areas showed scarce connective tissue with occasional lymphocytes. The scores of the microscopical hepatic lesions of the three infected groups (VAC1, VAC2, and IC) are summarized in Figure 1.

Fibrous perihepatitis consists of focal fibrosis with or without infiltrate of lymphocytes and plasma cells in the Glisson’s capsule coinciding with the healing of the lesions induced by migrating larvae when they penetrate in the liver or the healing of superficial migratory tracts (Figure 2a,c,e). The severity of this lesion was significantly lower (*p* < 0.05) in the VAC1 group compared with the IC group, while in theVAC2 group it was slightly lower than in the IC group but without significant differences (Figure 1). Chronic tracts are composed of fibrosis and macrophages loaded with hemosiderin pigment (Figure 2b,d,f) as a consequence of the healing migratory tunnels with hemorrhages induced by migrating larvae. The VAC1 group presented significantly lower chronic tracts (*p* < 0.05) than the IC group whereas VAC2 groups showed slightly lower severity of chronic tracts than IC group but with no significant differences (Figure 1).

Bile duct hyperplasia consists of enlargement of bile ducts and presence of papilla to increase the surface of epithelium (Figure 3a,c,e) and it is due to the presence of adult *F. hepatica* in the lumen of bile duct causing mechanical irritation with the spines and oral sucker and chemical irritation due to the excretory secretory (ES) product that the adult parasite release. The VAC1 group showed significantly (*p* < 0.01) lower severity bile duct hyperplasia than the IC group (Figure 1) while no significant differences were found between VAC2 and IC groups. Portal fibrosis is another common hepatic lesion of chronic fasciolosis and consist of fibrosis in the portal spaces, particularly in those showing bile duct hyperplasia (Figure 3a,c,e). The VAC1 group showed significantly (*p* < 0.01) lower portal fibrosis than the IC group (Figure 1) but no significant differences were recorded for portal fibrosis between the VAC2 and IC groups (Figure 1).

Granulomas composed of a necrotic center surrounded by a core of epithelioid macrophages and in some cases multinucleate giant cells and outer inflammatory infiltrate of eosinophils and lymphocytes (Figure 3b,d,f) were found in the two vaccinated groups and in the IC group, but the individual variability of this lesion was high in the three groups and there were no significant differences between vaccinated groups and the IC group (Figure 1). Inflammatory infiltrate of eosinophils was common in portal areas (Figure 4a,c,e), in the periphery of granulomas and surrounding parasite eggs that reached the hepatic parenchyma. The group VAC1 showed a significantly (*p* < 0.01) less severe infiltration of eosinophils than group IC, whereas the group VAC2 presented highly individual variability and a slightly lower infiltration than group IC but without significant differences (Figure 1). Inflammatory infiltrate of lymphocytes and plasma cells was also common in portal areas (Figure 4a,c,e) and in the periphery of granulomas, in some cases this infiltrate was arranged forming lymphoid follicles. The group VAC1 showed a less severe (*p* < 0.05) infiltration of lymphocytes and plasma cells than group IC, among the group VAC2 and IC no significant differences were recorded although the infiltration of eosinophils tends to be more pronounced in the group IC (Figure 1). Globule leukocytes with wide cytoplasm containing large eosinophilic granules were observed in the epithelium of some bile ducts, particularly in hyperplastic ducts. The score for globule leukocytes was similar in the three infected groups without significant differences between them (Figure 1). *F. hepatica* eggs were observed within some bile ducts (Figure 4b,d,f) and in some cases in the hepatic parenchyma associated to a severe inflammatory response with abundant eosinophils. The score for presence of parasite eggs was higher in the IC group than in VAC1 group, without significant differences between the IC and VAC2 groups (Figure 1).

### 3.2. Evaluation of F. Hepatica Morphology

During chronic fasciolosis the parasites are allocated within the bile ducts as adults and as consequence of their hermaphrodite reproductive strategy they are able to shed thousands of eggs per day. In this study conducted at 15 wpi, *F. hepatica* adults were allocated within the hyperplasic bile ducts surrounded by a thick fibrous capsule (Figure 5a). The anatomy of these adults showed the gastrointestinal and reproductive tracts intact with the normal morphology (Figure 5b,c) as has been described previously [13,14]. Moreover, the tegument presented well-developed eosinophilic spines (Figure 5b) which cause mechanical damage of the bile duct epithelium. These adults were reproductively viable confirmed by the high number of eggs found within the reproductive tract of the parasites observed and the eggs found within the lumen of the bile ducts (Figure 5c). Apparently, these eggs were viable containing cells which correspond to the fertilized ovum and vitelline cells enclosed in a yellowish capsule.

Unexpectedly, four sheep from the VAC1 group showed occasional degenerated forms of *F. hepatica* (Figure 5d–f) within bile ducts that were surrounded by a thick band of fibrous tissue and by an unusual severe outermost granulomatous inflammatory infiltrate with the presence of lymphoid follicles (Figure 5d). The majority of epithelium of bile ducts containing degenerated flukes was desquamated but it was possible to identify some epithelial bile duct cells using the cytokeratin AE1/AE3 antibody as marker for this cell type (Figure 6d). Occasionally, necrotic tissue was found surrounding degenerated parasites. Compared with the viable parasites, degenerated flukes showed a marked homogeneous eosinophilia lacking cell nuclei. Moreover, the tegument displayed a loss of spines and its morphology changes drastically toward a waving structure shedding acidophilic tissue associated and the reproductive and digestive systems could not be identified (Figure 5e). Additionally, it was also possible to observe degenerated eggs which could be identified in some cases according to their morphological features and the sclerotin-based eggshell (Figure 5f). These degenerated flukes were considered dead flukes and the bile ducts containing them were associated to a severe granulomatous reaction characterized by a layer of large epithelioid macrophages and multinucleate giant cells surrounding the dead flukes and necrotic tissue, a middle layer mainly composed of lymphocytes with lymphoid follicle formation, some of them showing large germinal center, and an outer layer of fibrous connective tissue.

The inflammatory infiltrates associated to degenerated flukes from the group VAC1 were characterized by immunohistochemistry using primary antibodies described in Table 1 and scored in base of the number of positive immunostained cells (Table 2). The immunohistochemical study revealed that the inflammatory infiltrates surrounding degenerated flukes from in the VAC1 group contained a high number of macrophages alternatively activated-M2 (CD163+) (Figure 7a), as well as a high number of macrophages and multinucleate giant cells strongly labelling by the lysozyme and S100 antibodies (Figure 7b,c, respectively). The CD68 antibody was expressed weakly by epithelioid macrophages and multinucleate giant cells in the inner layer of granulomas while it was expressed strongly by circulating monocytes, Kupffer cells, and macrophages located at the periphery of granulomas and in portal spaces. CD163+ cells were also present in the outermost inflammatory infiltrate of granulomas but they were not found within the lymphoid follicles. The CD163 stain is strictly membranous displaying a strong intensity. Thus, the morphology of the CD163+ stained cells are compatible with macrophages. However, lysozyme and S100 antibodies performed a strong cytoplasmic stain that immunolabeled the epithelioid macrophages highlighting the presence of multinucleated giant cells. They were also located in the middle area between the fibrous tissue and the outermost inflammatory infiltrate with no positivity within the lymphoid follicles.

The outer inflammatory infiltrate was composed by abundant lymphocytes, the majority of them reacted with the CD3 antibody (Figure 7d), whereas the presence of B cells (IgG lambda light chain+) was moderate (Figure 6e) and the FoxP3+-expressing regulatory T cells was scarce (Figure 7f). The cells CD3+ and B cells were localized in larger quantities in the inflammatory infiltrates outside the lymphoid follicles being present to a lesser extent within them. The staining pattern of both CD3 and IgG-lambda light chain antibodies was cytoplasmic with membranous strong intensity showing a morphology fully compatible with lymphocytes. The Foxp3+ cells showed nuclear and cytoplasmic staining pattern with a strong intensity, being localized dispersed within the outermost inflammatory infiltrate. Moreover, these infiltrates presented a high number of antigen presenting cells positive to MHC-II (HLA-DR+) marker (Figure 6a) displaying a cytoplasmic stain with moderate intensity and morphologically compatible with lymphocyte and macrophages. This positivity was present mainly within the lymphoid follicles but also dispersed along the inflammatory infiltrate area. However, the number of mature dendritic cells (CD208+) (Figure 6b) and proliferating cells (Ki-67+) (Figure 6c) was very scarce. The CD208 positivity was membranous with moderate intensity immunolabelling a very low number of cells in the outermost inflammatory infiltrate localized mainly in the surroundings of the lymphoid follicles. The Ki-67 marker was intranuclear with a strong intensity being localized randomly dispersed within the outermost inflammatory infiltrate.

## 4. Discussion

The VAC1 included in the present study was considered partially protective since it induced a fluke burden reduction of 37.2% (*p* = 0.002) and a significant reduction (*p* = 0.03) of gross hepatic lesion compared to the IC group. However, the VAC2 was considered non protective since it did not induce significant reduction of fluke burdens and gross hepatic lesions with respect to the IC group [11]. The results of the present study confirm the protective nature of VAC1 in term of histopathological hepatic lesions since it induced significant reduction of the majority of microscopical lesions scored (fibrous perihepatitis, chronic tracts, bile duct hyperplasia, portal fibrosis, inflammatory infiltrate of eosinophils, lymphocytes and plasma cells and presence of parasite eggs in bile ducts and hepatic parenchyma) in the VAC1 group with respect to the IC group. The only two lesions evaluated that showed no significant differences between the VAC1 group and the IC group were granulomas and infiltrate of globule leukocytes. On the other hand, the results of the present study also confirm that VAC2 was non protective since none of the microscopical hepatic lesions evaluated showed significant differences between the VAC2 and the IC groups. Among a variety of hematological, serological, and immunological parameters, the hepatic damage score is considered the best single index indicator of fluke burden, as has been previously reported in concern to *F. gigantica* by [15]. However, not much attention has been paid to evaluating hepatic damage as an additional indicator of protection in vaccine trials and only some studies have taken it into consideration [9,11,16,17,18,19].

Apart from the reduction of the above mentioned histopathological hepatic lesions, liver samples from the VAC1 group showed degenerated adults *F. hepatica* specimens within bile ducts, surrounded by a severe granulomatous inflammatory response. Degenerated flukes could be easily distinguished from viable flukes because they did not display the normal internal structures including reproductive and digestive tracts described previously [13,20,21,22,23] and, conversely, the internal body presented a homogenous acidophilic material that could be compatible with proteinaceous material after the denaturation of the proteins. Degenerated flukes have not been reported in previous pathological studies in experimental vaccine trials against *F. hepatica* [11,16,17,19,24] or in experimental *F. hepatica* infections in sheep [25,26] and goats [27]. Furthermore, in rats the effective host response against *F. hepatica* is supposed to occur during early stages of infection at the intestinal lamina propria or peritoneal cavity while the lumen of the bile ducts where adult flukes are located is considered a relatively immunologically safe environment [28,29]. It has been reported that some migrating flukes may die during the migratory stage and become encapsulated in the hepatic parenchyma surrounded by a fibrous connective tissue capsule forming a cyst that may caseate and become mineralized [30]. In the present study degenerated flukes were found within enlarged bile ducts, moreover, some of them showed degenerated eggs which confirm they had reached maturity before they died. This is the first study reporting degeneration of adult flukes within bile ducts in a vaccine trial for *F. hepatica*. This finding may explain the lower reduction of fecal egg counts (FEC) than fluke burdens in the VAC1 group, since the degenerated flukes contained degenerated eggs, they could have been producing viable eggs during some weeks and became degenerated later.

The mechanism implicated in the occurrence of this aberrant morphology of the adult parasites after vaccination is certainly unknown, but they were only observed in the VAC1 group, which also showed a significantly fluke burden reduction with respect to the infected control group, while VAC2 in which degenerated flukes were not found, did not induce significant fluke burden reduction with respect to the IC group. It is possible that the host immune response generated by VAC1 may have induced an effect on the survival of adult flukes causing the degeneration of some of them, or the blockage of some of the four proteins used in vaccine formulation (FhCL1, FhLAP, FhHDM, and FhPrx) could have affected the viability of adult flukes in the VAC1 group. The adjuvant used in the vaccine formulation may also have influenced this effect since the only difference between the two vaccinated groups was the adjuvant. The presence of a very severe granulomatous reaction with the organization of lymphoid follicles surrounding the fibrous tissue close to the lumen of the bile ducts containing dead flukes was similar to that reported in granulomas associated to disintegration in porcine cysticercosis [31], human neurocysticercosis [32], and in other parasitic infections such as schistomiasis or leishmaniasis [33] and in granulomatous cholangitis associated to the trematode *Campula* spp. in cetaceans [34]. The presence of a well-formed granuloma-like entity surrounding the degenerated adult flukes has not been described before in vaccination studies in fasciolosis.

The characterization of the local cellular immune response induced by the vaccine and/or the parasite infection may provide key information regarding the type of cell subsets involved in it and consequently, help to better comprehend the mechanism triggered during the pathogenesis of *F. hepatica* infection. In this study, the immunological characterization of this inflammatory infiltrates revealed the presence of epithelioid macrophages and multinucleate giant cells strongly expressing both S100 and lysozyme antibodies forming an inner band surrounding degenerated flukes and necrotic tissue. Epithelioid cells are modified tissue macrophages composing certain granulomas associated with intense immunological activity and that can be induced by several infectious agents [35,36,37]. Lysozyme and S100 antibodies that are specific of phagocytes are claimed to immunolabel the epithelioid macrophages as well [38,39]. Moreover, there was a very high number of CD163+ macrophages which corresponds to M2-macrophages possessing mainly a homeostatic anti-inflammatory and tissue-repair function after a severe tissue injury, as occurs in human or cattle [40,41,42,43,44]. In the present study, M2-macrophages may have played a role in the formation of the abundant connective band of the middle band of the granulomas.

CD3+ T lymphocytes constituted the major populations of the outer layer of the granulomas, this result is in agreement with the high number of this cell type in granulomas associated to remnants of the dead parasite *Hypoderma lineatum* but they were scarce in the periphery of viable larvae [45] and in the periphery of granulomatous cholangitis associated to the trematode *Campula* spp. in cetaceans [34]. This suggest that CD3+ T lymphocytes play a role in the local host response, either by producing cytokines to induce macrophage activation or tissue repair. B lymphocytes and plasma cells expressing IgG-lambda light chain were also abundant in the peripheral layer of the granulomas, suggesting a strong local humoral response, also reported in other parasitic granulomas [34,45].

The high number of HLA-DR+ cells located specifically within the lymphoid follicles which can be B cells, dendritic cells, and/or other antigen presenting cells, suggest that the role of these cell types is important in parasitic granulomas [33,46,47]. These HLA-DR+ cells are involved in the MHC class II-restricted antigen presentation that is essential for CD4+ T cell-dependent immune responses suggesting an efficient adaptive immune response against *F. hepatica* in the VAC1 group through an active presentation of antigens within the parasite-induced lymph follicles. Additionally, the low presence of CD208+ cells which is a marker expressed in human and ruminants by dendritic cells upon activation [43,48,49] indicates a less relevant role of this cell population at this location.

The low number of Treg cells (FoxP3+) in the granulomas associated to dead flukes contrasts with the expansion of FoxP3+ cells in the liver of experimentally infected sheep and goats, particularly at the periphery of enlarged bile ducts [50,51]. The expansion of this cell type in helminth infections is considered an important immunomodulatory effect to facilitate parasite survival [52]. It is possible that VAC1 has had some effect blocking the expansion of Foxp3 cells, which may have caused the local immune response to be more effective against flukes than in the IC and VAC2 groups. The low number of Ki-64+ cells in the granulomas indicates a low proliferation of cells at this late granuloma-like entity as has been reported by other authors in granulomatous lesions indicating the recruitment of cells mainly from blood instead of multiplying in situ [44].

## 5. Conclusions

The present study reports for the first time the presence of degenerated flukes associated to a severe granulomatous inflammation in bile ducts from sheep immunized with a partially protective vaccine (VAC1) composed of four recombinant proteins from *F. hepatica* in adjuvant Montanide 61 VG. The characterization of the granulomas associated to degenerated flukes revealed a high number of type 2 macrophages (CD163+) and CD3+ T lymphocytes with a low population of Foxp3+ Treg cells. Since sheep from the VAC1 group showed a significantly lower fluke burden and hepatic lesions than those from the infected control group, it is feasible that VAC1 may have induced certain effective host immune responses and at least, part of this response could have occurred against adult flukes allocated within the bile ducts.

## Figures and Tables

**Figure 1 animals-11-02869-f001:**
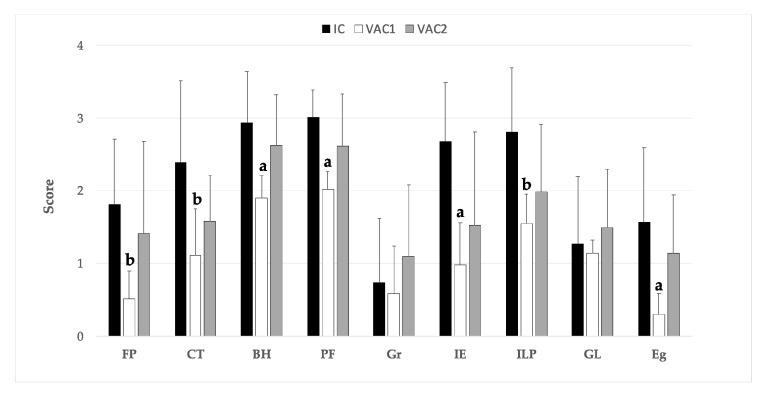
Evaluation of histopathological findings after *Fasciola hepatica* infections from immunized (VAC1 and VAC2) and unimmunized sheep (IC). Bars and error bars represent mean (*n* = 10) and SD, respectively. Score represents the severity of the lesions as: 0, absent; 1, mild; 2, moderate; 3, severe; 4, very severe. **^a,b^** Statistical difference (*p* < 0.01 and *p* < 0.05, respectively) compared to the IC group. FP: fibrous perihepatitis; CT: chronic tracts; BH: bile duct hyperplasia; PF: portal fibrosis; Gr: granulomas; IE: inflammatory infiltrate of eosinophils; ILP: inflammatory infiltrate of lymphocytes and plasma cells; GL: globule leukocytes; Eg: *F. hepatica* eggs.

**Figure 2 animals-11-02869-f002:**
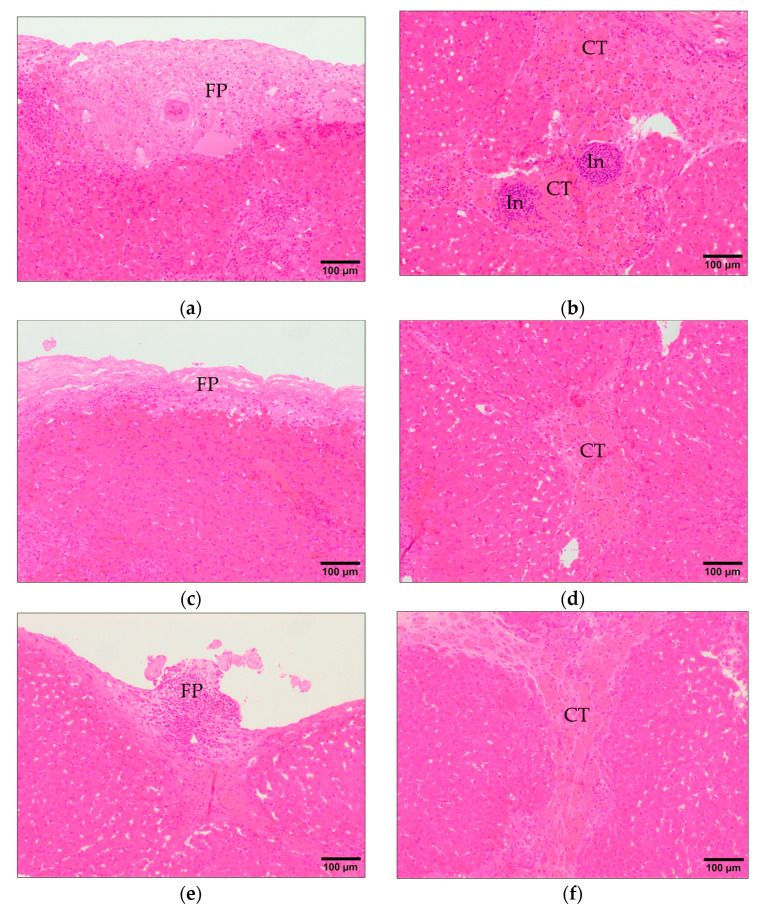
Main histopathological findings related to fasciolosis. (**a**,**c**,**e**) Fibrous perihepatitis (FP) in the Glisson’s capsule with focal fibrosis and inflammatory infiltrate from IC, VAC1, and VAC2 groups, respectively; (**b**,**d**,**f**) chronic tracts (CT) displaying fibrosis and macrophages loaded with hemosiderin pigment from IC, VAC1, and VAC2 groups, respectively. In: inflammatory infiltrates. H&E stain.

**Figure 3 animals-11-02869-f003:**
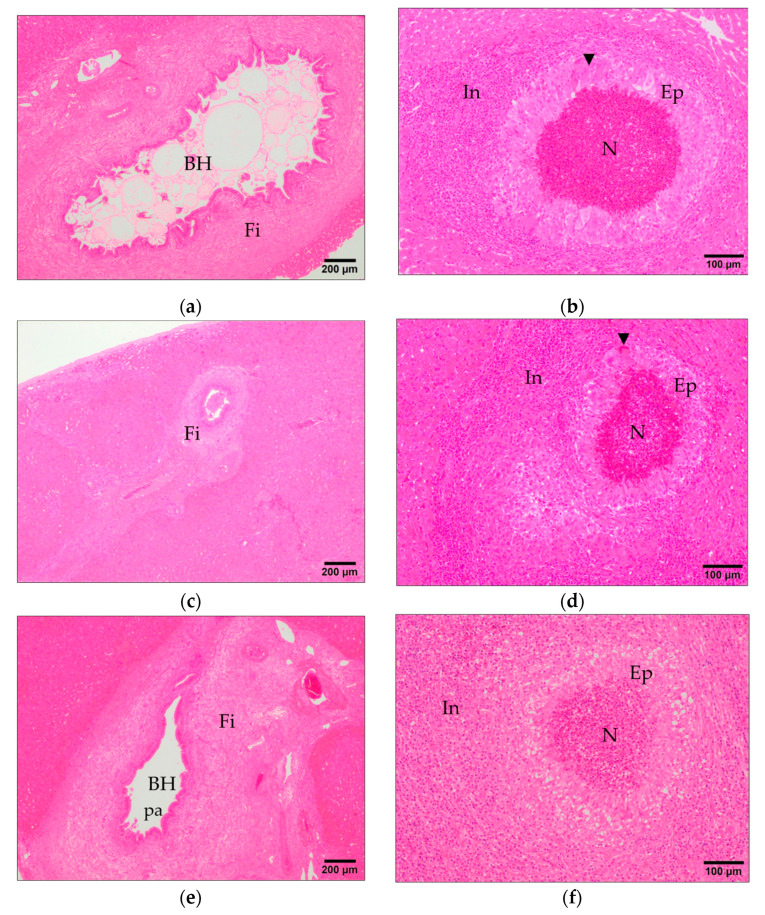
Main histopathological findings related to fasciolosis. (**a**,**c**,**e**) Portal fibrosis showing a thick fibrosis band (Fi) surrounding a hyperplasic bile duct (BH) consisting of enlargement of bile ducts and presence of papilla (pa) to increase the surface of epithelium from IC, VAC1, and VAC2 groups, respectively. (**b**,**d**,**f**) Granuloma with a necrotic center (N) surrounded by a core of epithelioid macrophages (Ep) and in some cases multinucleate giant cells (arrowhead) and outer inflammatory infiltrates (In) from IC, VAC1, and VAC2 groups, respectively. H&E stain.

**Figure 4 animals-11-02869-f004:**
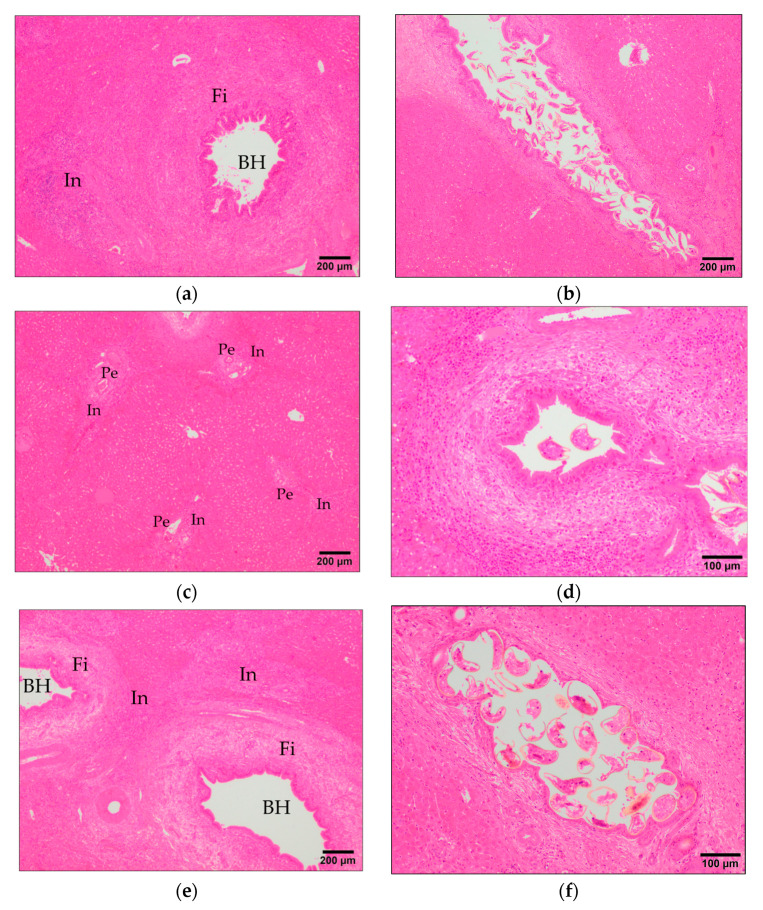
Main histopathological findings related to fasciolosis. (**a**,**c**,**e**) Inflammatory infiltrates (In) of eosinophils and/or lymphocytes and plasma cells from IC, VAC1, and VAC2 groups, respectively; (**b**,**d**,**f**) Eggs contained within bile ducts from IC, VAC1, and VAC2 groups, respectively. BH: bile duct hyperplasic; Fi: fibrosis band; Pe: periportal space. H&E stain.

**Figure 5 animals-11-02869-f005:**
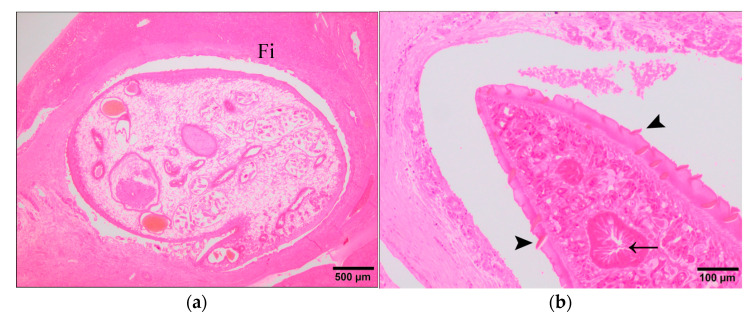
Anatomical morphology of *Fasciola hepatica*. (**a**–**c**) IC group; (**d**–**f**) VAC1 group. Fi: fibrous tissue; Lf: lymphoid follicles; Arrowhead: tegument spines of a normal *Fasciola hepatica*; Arrow: gastrointestinal tract of a normal *Fasciola hepatica*; *: viable reproductive tract with eggs of a normal *Fasciola hepatica*; DF: degenerated adult of *Fasciola hepatica*; In: inflammatory infiltrate; Dt: degenerated tegument; Arrowhead: *F. hepatica* eggs. H&E stain.

**Figure 6 animals-11-02869-f006:**
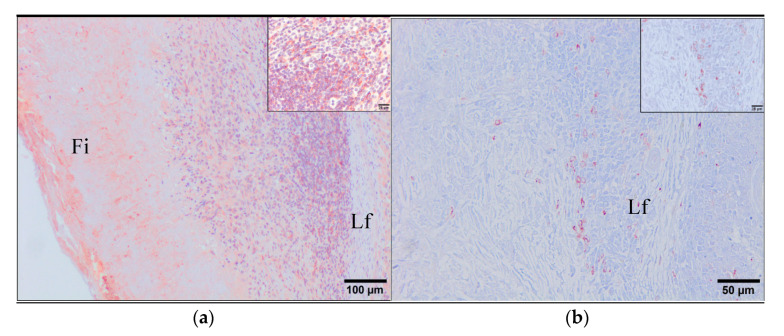
Immunohistochemical characterization of the inflammatory infiltrates surrounding the degenerated adults of *F. hepatica* in immunized sheep with a multivalent vaccine and Montanide 61 VG as adjuvant. The markers used were: (**a**) MHC-II+ antigen presenting cells; (**b**) CD208+ mature dendritic cells; (**c**) Ki67+ proliferating cells; (**d**) AE3/AE1+ epithelial cells from a hyperplasic bile duct. Fi: fibrous tissue; Lf: lymphoid follicles; BH: hyperplasic bile duct; In: inflammatory infiltrate. Inner magnification showing the stain pattern. Positive immunostaining cells are shown in brown-red. ABC method.

**Figure 7 animals-11-02869-f007:**
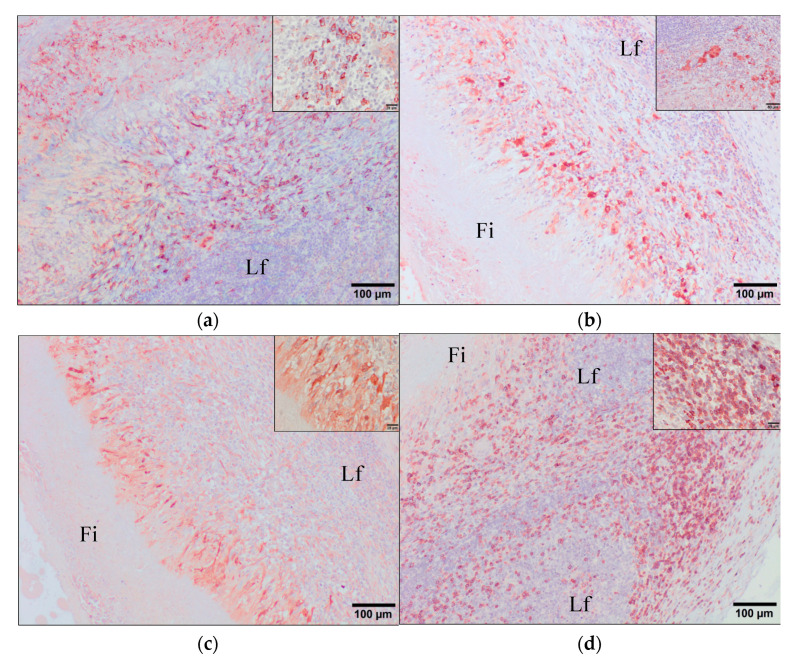
Immunohistochemical characterization of the inflammatory infiltrates surrounding the degenerated adults of *F. hepatica* in immunized sheep with a multivalent vaccine and Montanide 61 VG as adjuvant. Immunohistochemistry was carried out by using several markers for macrophages: (**a**) CD163; (**b**) lysozyme (inner magnification showing immunolabelled giant cells); (**c**) S100; and for lymphocytes: (**d**) CD3; (**e**) lambda light chains-B cells; (**f**) FoxP3-Treg. Fi: fibrous tissue; Lf: lymphoid follicles. Inner magnification showing the stain pattern. Positive immunostaining cells are shown in brown-red. ABC method.

**Table 1 animals-11-02869-t001:** Specificity of primary antibodies for immunohistochemical study.

Antibody ^1^, Clon or Ref, Source	Expression	Dilution	Retrieval Method
CD163 mAb, EdHu-1, BioRad	Mac. (M2)	1:400	TC pH6 Autocl. ^2^
Lysozyme pAb, A0099, Dako	Mac.	1:200	TC pH6 MW ^3^
S100 pAb, Z0311, Dako	Mac., DC	1:200	None
CD68 mAb, EBM11, Dako	Mac.	1:50	Pronase ^4^
CD3 pAb, A0452, Dako	Pan T lymp.	1:200	TC pH6 MW ^3^
Lambda light chains pAb, A0193, Dako	B cells	1:1000	Pronase ^4^
FoxP3 mAb, FJK-16s, eBiosciences	T reg	1:100	TC pH6 Autocl. ^5^
HLA-DR mAb, TAL.1B5, Dako	MHC-II APC	1:50	TC pH6 MW ^3^
CD208 (DC-LAMP) mAb, 1010E1.01, Dendritics	Mature DC	1:100	TC pH6 Autocl. ^2^
Ki67 mAb, MIB-1, Dako	Proliferating cells	1:100	TC pH6 Autocl. ^2^
Cytokeratin mAb, AE1/AE3, Dako	Epithelial cells	1:50	TC pH6 Autocl. ^2^

^1^ Monoclonal antibody (mAb) or polyclonal antibody (pAb). ^2^ Incubation with 0.1 M citric acid (pH 6), autoclave at 121 °C for 20 min. ^3^ Incubation with 0.1 M citric acid (pH 6), microwave for 10 min at sub-boiling temperature. ^4^ Incubation with 0.1% protease type XIV (Sigma-Aldrich) in 0.01 M PBS, pH 7.2 for 5 min at RT. ^5^ Incubation with 0.1 M citric acid (pH 6), autoclave at 135 °C for 10 min. Mac.: macrophages; DC: dendritic cells; lymp.: lymphocytes; Treg: regulatory T cells; APC: antigen presenting cells.

**Table 2 animals-11-02869-t002:** Evaluation of the immunostaining cells in the granulomatous lesions associated to dead flukes of *Fasciola hepatica*.

Antibody	Cellular Expression	Score ^1^
CD163	Mac. (M2)	Severe
Lysozyme	Mac.	Severe
S100	Mac., DC	Severe
CD68	Mac.	Mild
CD3	Pan T lymp.	Very severe
Lambda light chains	B cells	Moderate
FoxP3	T reg	Mild
HLA-DR	MHC-II APC	Moderate
CD208 (DC-LAMP)	Mature DC	Mild
Ki6	Proliferating cells	Mild
Cytokeratin (AE1/AE3)	Epithelial cells	Mild

^1^ Mild: <10 immunostained cells per field; moderate: 10–30 immunostained cells per field; severe: 30–50 immunostained cells per field; and very severe: >50 immunostained cells per field. Mac.: macrophages; DC: dendritic cells; lymp.: lymphocytes; Treg: regulatory T cells; APC: antigen presenting cells.

## Data Availability

The data presented in this study are available on request from the corresponding author.

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
