# Peer review of "A Partially Protective Vaccine for Fasciola hepatica Induced Degeneration of Adult Flukes Associated to a Severe Granulomatous Reaction in Sheep"

_animals, 2021, doi:10.3390/ani11102869_

Round 1

Reviewer 1 Report

I would like to thank the authors for addressing the previous comments. I just have some minor “cosmetic” and grammatical points that were overlooked in the previous version or newly introduced by the revision:

Line 29 in the abstract: „involved in the dead of adult flukes of F. hepatica. “ should be “in the death of”

Line 66 ff: “However, the growing resistance of the parasite to the chemical products and the concerns about chemical residues in food with their detrimental impact on the environment make critical the development of novel strategies more effective and sustainable.”

The sentence structure is poor, I would propose sth. like “… on the environment make the development of novel strategies critical that are more effective and sustainable.”

Line 417: the abbreviation FEC needs to be introduced or simply spelled out in full (makes sense for this non-parasitology journal)

Fig 5:

The figure legends says there should be a sub-figure g and h, which is not the case: “(d-h) VAC1 group”. It should be (d-f) instead.

Author Response

Thank you for your valuable comments.

We have corrected the manuscript according to the suggestions and comments.

Reviewer 2 Report

I have completed my evaluation of your manuscript and I found authors have addressed all the concerns raised in the previous version of the manuscript and the quality has improved after incorporating required modifications. Therefore, the manuscript may be considered for publication in this Journal.

Author Response

Thank you for your revision.

This manuscript is a resubmission of an earlier submission. The following is a list of the peer review reports and author responses from that submission.

Round 1

Reviewer 1 Report

The manuscript entitled "A partially protective vaccine for Fasciola hepatica induced dead of adult flukes associated to a severe granulomatous reaction in sheep". Title, abstract and overall rationale of work to some extent is excellent. However, there are some minor concerns, which needs to be addressed and needs minor revision.

1) Author must be write name of parasite in italic throughout the manuscript.

2)  Results are significant after given 2 doses of the vaccine and also parasite is died. I suggest, author, 3 doses of vaccine might be better and give most promising result in the future study.

3) Why author is not check the presence of Fasciola hepatica egg in the fecal sample.

4) Author also need to explain why this vaccine is not work in the Fasciola hepatica eggs.  

Author Response

The manuscript entitled "A partially protective vaccine for Fasciola hepatica induced dead of adult flukes associated to a severe granulomatous reaction in sheep". Title, abstract and overall rationale of work to some extent is excellent. However, there are some minor concerns, which needs to be addressed and needs minor revision.

We thank the reviewer for their positive comments about the manuscript.

1) Author must be write name of parasite in italic throughout the manuscript.

We thank the reviewer about this comment. We have checked and correct the name of the parasite throughout the manuscript for the italic format.

2)  Results are significant after given 2 doses of the vaccine and also parasite is died. I suggest, author, 3 doses of vaccine might be better and give most promising result in the future study.

We thank the reviewer about this recommendation. We have discussed vaccine protocols with several other groups working in F. hepatica vaccine development, we have decided to include two doses because for a commercial purpose three doses and several antigens may not be economically viable, particularly for sheep. In future trials we can consider using three doses depending on the number of antigens in the vaccine formula.

3) Why author is not check the presence of Fasciola hepatica egg in the fecal sample.

We thank the reviewer about this comment. The Fasciola hepatica eggs has been counted in the fecal samples and published previously by Zafra et al., 2021. Vet Res. 52(1):13.  doi: 10.1186/s13567-021-00895-0.

4) Author also need to explain why this vaccine is not work in the Fasciola hepatica eggs.  

We thank the reviewer about this comment. Although the focus of the present study was not the analysis of fluke burdens and FEC which were discussed in the previous paper (Zafra et al., 2021 Vet Res. 52(1):13.  doi: 10.1186/s13567-021-00895-0.). We think that the death of adult flukes may be one reason explaining the lower reduction of FEC than fluke burdens since some of the degenerated death flukes showed abundant degenerated eggs. Thus, they have been producing eggs until they died, and since FEC includes the number of eggs in feaces since week 8 post-infection to week 15 post-infection, parasites that were degenerated or dead by week 15 may have been producing eggs for several weeks. This has been mentioned in the revised manuscript (lines 701-704).

Reviewer 2 Report

The manuscript by Molina-Hernandez et al. addressed the question in how far histopathological parameters in sheep liver differ in sheep vaccinated against the liver fluke Fasciola hepatica compared to infected, non-vaccinated animals. The authors characterised the inflammatory infiltrates surrounding degenerated flukes with immunohistochemistry and found that vaccination significantly reduced the severity of several pathological parameters (e.g. fibrous perihepatitis, bile duct hyperplasia, portal fibrosis, inflammatory infiltrate of eosinophils and lymphocytes). In addition, the authors state that this is the first study reporting dead adult flukes within bile ducts in a vaccine trial for F. hepatica. With their results, the author conclude that a reduction in the liver damage could be helpful to evaluate vaccine efficacy against F. hepatica.

The study appears well conducted, includes necessary controls, and is largely well written (exceptions see below). However, I have some major and minor issues, especially with the lack of representative histological images and scoring data from all treatment groups, and the authors’ strong emphasis on the finding of dead worms (even in their manuscript title):

(1) As I understand, one of the aims of the manuscript is to promote histopathological scoring as a readout parameter in vaccination trials. It is needed to depict in all relevant figures (Fig. 2, 3, 5, 6) representative images for both, the infected control group and the VAC1 group. Currently, only images for the VAC1 group are shown, which makes it impossible for the reader to judge on the strength of vaccine effects and the easiness or difficulty when assessing histological parameters in such trials. At least as supplementary data, images of the infected control should be added.

(2) Abstract: The last sentence states “This is the first study reporting the dead of adult F. hepatica parasites in a vaccine trial”. This should be rephrased to be more specific, similar to the phrases in the discussion where the statement is limited to the presence of dead adult flukes in bile ducts.

(3) Line 98f: When the authors introduce the two kinds of vaccine formulations in their introduction, they should give reference to their previous study that showed the protectivity and they should ideally mention some more background information (what does “partically protective” mean here, i.e. what was the percentage of fluke burden reduction).

(4) Table 2 gives scores that indicate the number of cells per viewing field of VAC1 group animals. How do these scores relate to infected and uninfected control animals? These data should be added. This is important to answer, e.g.: are Tregs and dendritic cells (that have only a low abundance) at all infiltrating after infection, or are they present at the same low level at steady state? Did infected animals have a higher score for Tregs compared to VAC1 animals (the authors speculate in their discussion about a blocking effect on Treg numbers by the vaccine). Which cell types are reduced in their number to which extent in VAC1 compared to infected animals? To answer all this and more, I would propose to add three columns with all relevant scores to table 2, one column for each group (VAC1, infected, uninfected).

(5) Line 260f: “…it was possible to identify some epithelial bile duct cells using the cytokeratin AE1/AE3 antibody as marker for this cell type” à give reference to a figure. Or are these “data not shown”?

(6) Line 270: “These degenerated flukes were considered dead flukes” à Can the authors include in the text how often (presumably) dead worms were found within tissue sections? If 6 hepatic samples were stained per animal, how many of these contained a fluke and how many of those appeared dead? And were flukes only considered dead based on a degenerated phenotype found in H&E images, or were (different liver lobes) collected, flukes dissected from bile ducts and then assessed for vitality? If the latter was not done, all conclusions on “dead adult worms after vaccination” need to be tuned down throughout the text, including the manuscript title and abstract. I know from in vitro exposure of flukes to drugs that they can appear extremely damaged and almost immotile, but they are in fact still alive.

(7) Line 350f: I believe this must be rephrased. Reference 15 is about an index for Fasciola gigantica, while the authors state the index was established for F. hepatica.

(8) Line 425: The authors conclude on a “non-relevant role” of dendritic cells because of their low abundance. I would rephrase this to “less relevant” or tune down the statement even further. A low abundance of immune cells per se does not mean they are not important (see ILCs, Tregs, Bregs etc.)

There are several issues with the figures and figure legends:

Legend Fig. 1: Please state whether the error bars represent SEM or SD and state the number of animals used to build the mean value (legend should contain all relevant details without need to check the methods section).

Legend Fig. 2. There are several issues: I read twice “Main histopathological findings related to fasciolosis”. Should one statement be removed? There are also two times “(a)”, while the introduced abbreviations “N”, “Ep” and “In” and the mentioned arrowhead are missing in the figures. Furthermore, the description of the subfigures within the legend does not seem to match the depicted images and the description within the text, e.g. Fig. 2b shows chronic tracts according to the text (line 189), but the legend says different.

Fig. 3 is missing. If I may guess, legend of Fig. 2 is actually the legend of Fig. 3.

Legend Fig. 4: Several abbreviations depicted in the images are not defined in the legend. It is not specified what arrows and asterisks actually indicate. Probably again a wrong legend for the figure.

Fig. 5 and 6: I would prefer to have an insert showing a magnification of some positively stained cells in each of the images. Descriptions in the text such as for cells “displaying a cytoplasmic stain with moderate intensity and morphologically compatible with lymphocyte and macrophages” are difficult to follow in the low-magnification overview images.

Language:

Title: dead > death

Line 22: zoonoses > zoonosis

Line 37: dead > death

Lines 57/58: parasite enters … where they … and starts à is a mixture of singular and plural, please harmonize

Lines 64 ff: sentence is incomplete

Line 82: have been described > have described

Line 232 and 247, 257: write “F. hepatica” in italics

Line 250: “4b y 4c”, replace “y” with “and”

Line 285: “multinucleate giant cells strongly expressing the lysozyme and S100 antibodies” > these cells do not express antibodies, but the antibodies revealed expression of these proteins by the cells; please rephrase

Line 364: to occurs > to occur

Line 443: “a … fluke burdens” > singular or plural

Author Response

The manuscript by Molina-Hernandez et al. addressed the question in how far histopathological parameters in sheep liver differ in sheep vaccinated against the liver fluke Fasciola hepaticacompared to infected, non-vaccinated animals. The authors characterised the inflammatory infiltrates surrounding degenerated flukes with immunohistochemistry and found that vaccination significantly reduced the severity of several pathological parameters (e.g. fibrous perihepatitis, bile duct hyperplasia, portal fibrosis, inflammatory infiltrate of eosinophils and lymphocytes). In addition, the authors state that this is the first study reporting dead adult flukes within bile ducts in a vaccine trial for F. hepatica. With their results, the author conclude that a reduction in the liver damage could be helpful to evaluate vaccine efficacy against F. hepatica.

The study appears well conducted, includes necessary controls, and is largely well written (exceptions see below). However, I have some major and minor issues, especially with the lack of representative histological images and scoring data from all treatment groups, and the authors’ strong emphasis on the finding of dead worms (even in their manuscript title):

We thank the reviewer for their summary, and we are pleased to answer their major and minor issues to improve the manuscript.

(1) As I understand, one of the aims of the manuscript is to promote histopathological scoring as a readout parameter in vaccination trials. It is needed to depict in all relevant figures (Fig. 2, 3, 5, 6) representative images for both, the infected control group and the VAC1 group. Currently, only images for the VAC1 group are shown, which makes it impossible for the reader to judge on the strength of vaccine effects and the easiness or difficulty when assessing histological parameters in such trials. At least as supplementary data, images of the infected control should be added.

We thank the reviewer for their comments. Figure 2 has been replaced by Figures 2, 3 and 4 showing more pictures related to the main histopathological parameters found in the IC, VAC1 and VAC2 to compare among groups. Figure 5 is related to the morphological changes observed between the IC and VAC1 groups.The Figures 6 and 7 are related to the immunohistochemical characterization of the severe granulomatous inflammatory infiltrates found only in some animals from the VAC1 group.

(2) Abstract: The last sentence states “This is the first study reporting the dead of adult F. hepatica parasites in a vaccine trial”. This should be rephrased to be more specific, similar to the phrases in the discussion where the statement is limited to the presence of dead adult flukes in bile ducts.

We thank the reviewer for this suggestion, and we changed the sentence mentioning the presence of dead flukes associated to a severe granulomatous inflammation in bile ducts in the abstract (line 45) and in the simple summary (lines 29-30).

(3) Line 98f: When the authors introduce the two kinds of vaccine formulations in their introduction, they should give reference to their previous study that showed the protectivity and they should ideally mention some more background information (what does “partically protective” mean here, i.e. what was the percentage of fluke burden reduction).

We thank the reviewer for this comment. To answer this issue, we introduced the sentence “The formulation of the two vaccines assessed in this study were published previously by [11] finding a reduction of fluke burden (37.2%) and egg output (28.71%) in comparison to IC group.” in the lines 124-126.

(4) Table 2 gives scores that indicate the number of cells per viewing field of VAC1 group animals. How do these scores relate to infected and uninfected control animals? These data should be added. This is important to answer, e.g.: are Tregs and dendritic cells (that have only a low abundance) at all infiltrating after infection, or are they present at the same low level at steady state? Did infected animals have a higher score for Tregs compared to VAC1 animals (the authors speculate in their discussion about a blocking effect on Treg numbers by the vaccine). Which cell types are reduced in their number to which extent in VAC1 compared to infected animals? To answer all this and more, I would propose to add three columns with all relevant scores to table 2, one column for each group (VAC1, infected, uninfected).

We thank the reviewer for this suggestion. The aim of the immunohistochemical study was to characterize the inflammatory infiltrate of granulomatous lesions associated to parasite debris only, this is why as is described in the lines 398-401, only four sheep from the VAC1 group showed degenerated forms of F. hepatica within bile ducts that were surrounded by a thick band of fibrous tissue and by an unusual severe outermost granulomatous inflammatory infiltrate with the presence of lymphoid follicles contrasting with the IC and VAC2 groups. Thus, the adult forms found in the VAC 2 and IC groups did not show these granulomatous inflammatory infiltrates and they presented only the typical fibrous band surrounding the bile ducts. Table 2 represents the immunohistochemical characterization and evaluation of these granulomatous lesions associated specifically to degenerated flukes of Fasciola hepatica found only in the VAC1 group that is why we only added the VAC1 group.

We can consider to characterize the portal inflammatory infiltrates of all animals, but due to the number of animals and antibodies and the time requires for laboratory work and cell counting, we will require at least one month to do this work.  

(5) Line 260f: “…it was possible to identify some epithelial bile duct cells using the cytokeratin AE1/AE3 antibody as marker for this cell type” à give reference to a figure. Or are these “data not shown”?

We thank the reviewer for this comment. We have added an image of the positivity obtained with the AE1/AE3 antibody in the figure 6 to confirm the location of the dead adults within the hyperplasic bile ducts. We added “(Figure 7d)” at the end of the sentence in line 404.

(6) Line 270: “These degenerated flukes were considered dead flukes” à Can the authors include in the text how often (presumably) dead worms were found within tissue sections? If 6 hepatic samples were stained per animal, how many of these contained a fluke and how many of those appeared dead? And were flukes only considered dead based on a degenerated phenotype found in H&E images, or were (different liver lobes) collected, flukes dissected from bile ducts and then assessed for vitality? If the latter was not done, all conclusions on “dead adult worms after vaccination” need to be tuned down throughout the text, including the manuscript title and abstract. I know from in vitro exposure of flukes to drugs that they can appear extremely damaged and almost immotile, but they are in fact still alive.

We thank the reviewer for this comment. The number of dead adults found from the VAC1 group were 1 or 2 per animal from 4 animals out of 10 sheep of this group. It has been stated “occasional” in line 398. We have identified degenerated flukes based on histological features and only 6 samples were collected per animal, this method is not appropriate for evaluating the number of dead flukes per animal, but it can be useful to identify dead or degenerated flukes in a group. We have used the term “degenerated” fluked instead of “dead” flukes in the revised manuscript, although the histological features (absence of nuclei in parasite cells, loss of histological structures such as digestive and reproductive tracts and intense eosinophilia are not compatible with a viable tissue of organism as all pathologists know and it can be interpreted as severely autolytic changes or coagulative necrosis. The results of the histological study were obtained after the fluke burden study was carried out, this is why fluke viability was not assessed in fluke burdens, but we will consider to do it in future trials.

(7) Line 350f: I believe this must be rephrased. Reference 15 is about an index for Fasciola gigantica, while the authors state the index was established for F. hepatica.

We thank the reviewer for this comment. In order to answer this issue and to be accurate with the citation which is related to F. gigantica as the reviewer indicate, the sentence has been changed to “Among a variety of haematological, serological and immunological parameters, the hepatic damage score is considered the best single index indicator of fluke burden, as has been previously reported in concern to F. gigantica by [15].” being in the lines 682-683.

(8) Line 425: The authors conclude on a “non-relevant role” of dendritic cells because of their low abundance. I would rephrase this to “less relevant” or tune down the statement even further. A low abundance of immune cells per se does not mean they are not important (see ILCs, Tregs, Bregs etc.)

We thank the reviewer for this comment, and we changed the sentence including “less relevant role” instead of “non-relevant role” in the line 763.

There are several issues with the figures and figure legends:

Legend Fig. 1: Please state whether the error bars represent SEM or SD and state the number of animals used to build the mean value (legend should contain all relevant details without need to check the methods section).

We thank the reviewer for this comment, and we added the sentence “Bars and error bars represent mean (n=10) and SD, respectively.” in line 206-207.

Legend Fig. 2. There are several issues: I read twice “Main histopathological findings related to fasciolosis”. Should one statement be removed? There are also two times “(a)”, while the introduced abbreviations “N”, “Ep” and “In” and the mentioned arrowhead are missing in the figures. Furthermore, the description of the subfigures within the legend does not seem to match the depicted images and the description within the text, e.g. Fig. 2b shows chronic tracts according to the text (line 189), but the legend says different.

We thank the reviewer for this comment. It seems that the figures 2 and 3 were merged in this version of the manuscript. We mended this issue and introduced the proper legend to each figure, as well as we added the missed figure 3, although Figures 2 to 4 were changed based on the comments suggested by the Reviewer 2 at the point 1 of this cover letter.

Fig. 3 is missing. If I may guess, legend of Fig. 2 is actually the legend of Fig. 3.

We thank the reviewer for this comment. We mended this issue by adding new figures 2, 3 and 4.

Legend Fig. 4: Several abbreviations depicted in the images are not defined in the legend. It is not specified what arrows and asterisks actually indicate. Probably again a wrong legend for the figure.

We thank the reviewer for this comment. We mended this issue adding the correct legend for figure 4. Also, we introduced the figures again to adapting them to the right size of the format of the manuscript.

Fig. 5 and 6: I would prefer to have an insert showing a magnification of some positively stained cells in each of the images. Descriptions in the text such as for cells “displaying a cytoplasmic stain with moderate intensity and morphologically compatible with lymphocyte and macrophages” are difficult to follow in the low-magnification overview images.

We thank the reviewer for this suggestion. We added an inner magnification image for each image of the figures 6 and 7 to improve the visualization of the stain pattern for each antibody.

Language:

Title: dead > death

We thank the reviewer for this appreciation. Based on the suggestion made by the Reviewer 2 at the point 6 of this cover letter, and we changed the word dead by “degeneration” in the title.

Line 22: zoonoses > zoonosis

We thank the reviewer for this appreciation, and we changed the word zoonoses by zoonosis in the line 20.

Line 37: dead > death

We thank the reviewer for this appreciation, and we changed the word dead by degeneration in the line 37.

Lines 57/58: parasite enters … where they … and starts à is a mixture of singular and plural, please harmonize

We thank the reviewer for this appreciation, and we changed the sentence to “Lately, the parasites enter the bile ducts where they develop into their adult form and start releasing up to 20,000–24,000 eggs per fluke per day”. in lines 70-72.

Lines 64 ff: sentence is incomplete

We thank the reviewer for this appreciation and we changed the verb being by make to give sense to the sentence in line 79.

Line 82: have been described > have described

We thank the reviewer for this appreciation, and we changed the verb tense to “have described” in line 95.

Line 232 and 247, 257: write “F. hepatica” in italics

We thank the reviewer for this appreciation, and we included the italic format in these lines and checked the name of the parasite throughout the manuscript.

Line 250: “4b y 4c”, replace “y” with “and”

We thank the reviewer for this appreciation, and we changed “y” by “and”.

Line 285: “multinucleate giant cells strongly expressing the lysozyme and S100 antibodies” > these cells do not express antibodies, but the antibodies revealed expression of these proteins by the cells; please rephrase

We thank the reviewer for this appreciation, and we changed “expressing” by “labelling by” in line 487.

Line 364: to occurs > to occur

We thank the reviewer for this appreciation, and we changed “to occurs” by “to occur” in the line 690.

Line 443: “a … fluke burdens” > singular or plural

We thank the reviewer for this appreciation, and we wrote “fluke burden” in line 781.
